# Weight Gain Predicts Metabolic Syndrome among North Korean Refugees in South Korea

**DOI:** 10.3390/ijerph18168479

**Published:** 2021-08-11

**Authors:** Yoon Jung Kim, Yo Han Lee, Yun Jeong Lee, Kyeong Jin Kim, Sin Gon Kim

**Affiliations:** 1Division of Endocrinology and Metabolism, Department of Internal Medicine, College of Medicine, Hallym University, Chuncheon 24253, Korea; yjkim99@hallym.or.kr; 2Graduate School of Public Health, Ajou University, Suwon 16500, Korea; vionic@ajou.ac.kr; 3Department of Preventive Medicine and Public Health, Ajou University School of Medicine, Suwon 16500, Korea; 4Ajou Institute of Korean Unification and Health Care, Suwon 16500, Korea; 5Division of Endocrinology and Metabolism, Department of Internal Medicine, Anyang Sam Hospital, Anyang 14030, Korea; songhope2005@naver.com; 6Division of Endocrinology and Metabolism, Department of Internal Medicine, College of Medicine, Korea University, Seoul 02841, Korea; papaya0707@naver.com

**Keywords:** North Korean refugee, weight gain, metabolic syndrome, percentage body fat

## Abstract

Previous cross-sectional studies showed that immigrants from low-income to high-income countries have higher risk of cardiovascular disease and type 2 diabetes mellitus. We investigated the association between weight gain during the resettlement in South Korea and risk of metabolic syndrome (MetS) among North Korean refugees (NKRs) in this cross-sectional study. In total, 932 NKRs aged 20–80 years in South Korea voluntarily underwent health examination from 2008 to 2017. We compared the risk of MetS and its components between the weight gain group (gained ≥5 kg) and the non-weight gain group (gained <5 kg, maintained or lost body weight) during resettlement in South Korea after defection from North Korea. Multiple logistic regression analysis predicted odds ratio of MetS on the basis of weight change, adjusting for covariates and current body mass index (BMI). We also evaluated the difference in body composition of NKRs between two groups. The prevalence of MetS in the weight gain group was 26%, compared to 10% in the non-weight gain group (*p*-value < 0.001). The weight gain group had a two-fold higher risk of MetS than the non-weight gain group after adjusting for current BMI (odds ratio 1.875, *p*-value = 0.045). The prevalence of central obesity, impaired fasting glucose, elevated blood pressure, and hypertriglyceridemia were higher in the weight gain group than the non-weight gain group (36% vs. 12%, *p*-value < 0.001; 32% vs. 19%, *p*-value < 0.001; 34 vs. 25%, *p*-value = 0.008; 19% vs. 13%, *p*-value = 0.025, respectively). The analysis of body composition showed that the percentage of body fat in the weight gain group was higher than in the non-weight gain group, indicating increased fat mass rather than muscle mass in the weight gain group as their body weight increased during resettlement (33.4 ± 6.53% vs. 28.88 ± 7.40%, *p* < 0.005). Excess weight gain after defection from North Korea increased the risk of MetS among NKRs in South Korea. It is necessary to monitor weight change among NKRs and their effect on their metabolic health in the long term.

## 1. Introduction

As the prevalence of obesity has increased worldwide due to dietary change and sedentary lifestyles, the prevalence of metabolic syndrome (MetS) has also rapidly increased [1]. MetS could be a public health problem in that it could be a risk factor for diabetes mellitus and cardiovascular disease [2,3]. Obesity is a well-known risk factor for MetS in the general adult population [4,5,6]. Obesity is etiologically linked with chronic low-grade inflammatory conditions that contribute to the pathogenesis of metabolic dysfunction and MetS [7]. Previous studies have shown that weight gain had a significant relationship with MetS [8,9]. According to a previous prospective study, 4.5 kg of weight gain during a 13-year follow-up increased the risk of MetS by 23% [8]. Another cross-sectional study revealed that the changes of body mass index (BMI) in early adulthood (25–40 years) were strongly associated with increased risk of metabolic components, such as increased Hemoglobin A1c (Hb A1c) or low high-density lipoprotein (HDL) cholesterol level [9]. Cross-sectional studies among Japanese populations revealed that weight gain increased the risk of MetS, even in a non-obese population [10,11]. Hashimoto et al. showed that weight gain after age of 20 years increased the risk of MetS by twofold in non-obese individuals [10].

Immigrants from low-income countries experienced weight gain when they migrated to high-income countries due to westernized lifestyle and dietary changes [12,13]. Immigrant women from Turkey and Chile to Sweden had a higher risk of obesity due to change of dietary habits and low physical activity in a cross-sectional study [12]. Another longitudinal study among immigrant Chinese women in the United States (US) showed that increasing length of US residence was associated with increased energy density in diet and sugar intake [13]. Furthermore, previous studies showed that immigrants from low- to high-income countries had a higher risk of cardiovascular disease and type 2 diabetes mellitus than the host population [14,15]. A cross-sectional study of South Asian immigrants in the US showed that South Asians had coronary artery disease more commonly than Caucasians [15]. However, few studies have assessed how this weight change among immigrants affects their metabolic health during resettlement in host country [16].

The ethnic background of the population of the Korean Peninsula remained unchanged for 5000 years, but Korea was divided into North and South Korea after the Korean War. While South Korea has achieved rapid economic growth by introducing a free-market economy system, North Korea has experienced severe poverty due to collapse of communism and continuous natural disasters. According to World Health Organization (WHO, Geneva, Switzerland) survey data in 2008, the North Korean population aged 25–64 years had very low obesity prevalence, 4.2% in men and 4.6% in women. On the contrary, obesity prevalence in the same age group is 38.9% in South Korean men and 25.3% in women according to a 2008 Korean National Health and Nutrition Examination Survey [17,18]. Exhausted from poverty and hunger, many North Koreans have left North Korea and resettled in South Korea through transit countries [19,20]. The number of North Korean refugees (NKRs) in South Korea reached 33,000 in December 2019 [21]. We previously showed that NKRs experienced weight gain and approached the average BMI of South Korean population 10 years after defection from North Korea [22]. In addition, while NKRs still had a lower BMI than the South Korean population, the prevalence of MetS among NKRs was similar with that of the South Korean population [23].

In the present study, we investigated the association between weight gain and the risk of MetS among NKRs in South Korea. NKRs with more than three of the five components were classified as having MetS (central obesity, impaired fasting glucose, elevated blood pressure, hypertriglyceridemia, and low HDL cholesterol). We hypothesized that NKRs who gained weight more than 5kg would have a higher risk of MetS. We also examined the difference in body composition between the two groups of NKRs with and without 5 kg or more weight gain after defection from North Korea by bioelectrical impedance analysis (BIA).

## 2. Methods

### 2.1. Study Participants

This study was conducted in the phase 1 survey of the North Korean Refugee Health in South Korea (NORNS) study [24]. The NORNS study is composed of two phases. The phase 1 study is a cross-sectional survey, and the phase 2 study is a follow-up survey 3.5 years after the phase 1 study. The NORNS study has taken place at Korea University Anam Hospital since October 2008 and still ongoing. Hana center, a representative welfare center supported by the South Korean government for assisting NKRs’ resettlement in South Korea, posted a notice about the study on the Internet and by telephone once a month, and NKRs who have expressed their intention to participate can take part in the study. From October 2008 to March 2017, a total of 932 NKRs (men = 192, women = 740) above 20 years living in Seoul, South Korea, were recruited. The participant’s survey consisted of health questionnaire and medical examination. Medical examination comprised anthropometric assessment, blood pressure, biochemical measurement, and analysis of body composition through BIA. We have described the protocols and methods of the study in detail previously [24]. Of the 932 participants, we excluded those whom data of body weight, waist circumference (WC), or history of body weight were missing or unreliable. Finally, 799 NKRs (men = 162, women = 637) aged 20–80 years were analyzed. All participants provided written informed consent and the study was approved by the Institutional Review Board of Korea University Anam Hospital (approval number: Ed08023).

### 2.2. Measurements

#### 2.2.1. Sociodemographic Characteristics

We produced a 42-item health questionnaire based on the existing questionnaire of Korea National Health and Nutrition Examination Survey (KNHANES) that is composed of six domains (demographic and migration information, disease history, mental health status, health-related lifestyles, female reproductive health, and sociocultural adaptation) (Appendix A). In this study, we used the sociodemographic information including age, sex, health-related lifestyle factors, education level, occupation, current income in South Korea, and migration information (time of defection from North Korea, time of arrival in transit country and South Korea). Health-related lifestyle factors included smoking, frequent alcohol drinking (more than one bottle of alcohol per week), and regular exercise (vigorous physical activity more than 1 h per week). Higher income was defined as more than 10^6^ Korean won (KRW)/month, and higher education level was defined as above college graduate. Participants were asked to recall their body weights in North Korea, transit countries, and South Korea. For the completeness of the questionnaire, we had the NKR interview with a medical doctor who defected from North Korea.

#### 2.2.2. Anthropometric Measurement

The health examinations were conducted by licensed nurses and trained volunteer. Body weight was measured in light clothing and height was measured without shoes by automatic system (GL-150; G-Tech International, Seoul, South Korea). BMI was calculated as weight (kg) divided by squared height (m^2^). WC was measured at minimum abdominal girth, midpoint between the lowest rib and iliac crest by nurses [25]. Systolic and diastolic blood pressure were measured by an autonomic blood pressure monitor (TM-2655P; Biospace, Tokyo, Japan) on the arm of a seated participant who rested in a sitting position for 10 min before measurement and used the mean value in the analysis [26]. Participants underwent BIA (Inbody720, Biospace, Seoul, South Korea) while wearing light clothes to analyze body muscle mass, fat mass, and percentage of body fat.

#### 2.2.3. Biochemical Measurement

Blood samples were drawn in the morning after an overnight fasting. Serum total cholesterol, triglyceride, HDL cholesterol, low density lipoprotein (LDL) cholesterol, and liver enzyme were measured using enzymatic method (TBA 200-FR; Toshiba, Tokyo, Japan). Plasma glucose concentrations were measured by the glucose oxidase method. Serum insulin levels were measured by radioimmunoassay (Diasource, Nivelles, Belgium). The homeostasis model assessment–insulin resistance (HOMA–IR) was computed using following formula: fasting plasma glucose (mg/dL) × fasting serum insulin (μU/mL)/405 [27]. Insulin secretary function (HOMA–ß) was calculated with following formula: 360 × fasting insulin (μU/mL)/fasting glucose (mg/dL)–63 [27].

Metabolic syndrome was defined when more than three of the following criteria were met according to modified NCEP–ATP III guideline [28]: (1) WC ≥ 90 cm (men), ≥85 cm (women); (2) fasting glucose ≥ 100 mg/dL; (3) systolic blood pressure ≥ 130 mmHg or diastolic blood pressure ≥ 85 mmHg; (4) HDL cholesterol < 40 mg/dL (men), <50 mg/dL (women); (5) triglycerides ≥ 150 mg/dL. Participants who took medication for diabetes or hypertension were regarded as having met the criteria for elevated fasting glucose or elevated blood pressure.

#### 2.2.4. Classification of NKRs by Change in Weight after Defection from North Korea

To investigate the effect of weight gain after defection from North Korea on the risk of MetS, we classified NKRs into weight gain group (gained body weight more than 5 kg after defection from North Korea) and non-weight gain group (gained less than 5 kg, maintained or lost their body weight during resettlement in South Korea), and compared metabolic health between two groups. The cut-off value of 5 kg was chosen on the basis of previous large prospective cohort studies that investigated the association between weight change and coronary heart disease [29,30].

### 2.3. Statistical Analysis

Comparison of sociodemographic and metabolic characteristics in NKRs were presented by weight gain and non-weight gain group. Comparison of categorical variables were described as frequency (percentage) with Pearson χ^2^ test. Continuous variables were described as mean ± standard deviation using analysis of variance (ANOVA). Relationship between weight gain and risk of MetS was investigated with multivariate logistic regression analyses using MetS risk as a dependent variable. Odds ratio (OR) and 95% confidence intervals (95% CI) were calculated with non-weight gain group as the reference group. The logistic regression was conducted in two steps. In Model 1, the ORs were adjusted for age, sex, smoking, alcohol drinking, regular exercise, education, income, stay duration of transit countries, and defection period. In the second step (Model 2), the ORs were adjusted by adding the current BMI and lipid medication to covariates in Model 1. *p*-value < 0.05 was considered as significant. All statistical analyses were performed using SPSS 26.0 software (SPSS Inc., Chicago, IL, USA).

## 3. Results

### 3.1. Characteristics of The Study Population

The demographic characteristics of the NKRs according to weight gain after defection from North Korea are shown in Table 1. Among 799 participants, 253 (32%) were assigned to the weight gain group and 546 (68%) of NKRs to the non-weight gain group. The mean age and sex distribution were similar between the two groups. The weight gain group had higher mean value of body weight, BMI, and WC than the non-weight gain group. Mean fasting glucose, triglyceride, ALT levels, and blood pressure were higher in the weight gain group than non-weight gain group. Mean value of HOMA–IR was higher in weight gain group than non-weight gain group (1.68 ± 1.11 vs. 1.38 ± 1.2, *p*-value = 0.002). There was no significant difference in alcohol drinking, smoking, and physical activity between the two groups. Defection period from North Korea was longer in the weight gain group than non-weight gain group (7.91 ± 4.97 vs. 6.67 ± 4.62 years, *p*-value = 0.001). The weight gain group had a lower mean body weight and BMI in North Korea than the non-weight gain group, although they had a higher BMI and body weight than non-weight gain group on examination in South Korea. The mean weight change was 10 ± 4.8 kg in weight gain group and −0.5 ± 3.96 kg in non-weight gain group (*p*-value < 0.001).

### 3.2. Prevalence of Individual Metabolic Components in The Weight Gain and Non-Weight Gain Groups

Metabolic syndrome was 2.6 times more prevalent in the weight gain group than the non-weight gain group (26% vs. 10%, *p*-value < 0.001) (Table 2). The prevalence of central obesity in the weight gain group was three times higher than in non-weight gain group (36% vs. 12%, *p*-value < 0.001). Prevalence of impaired fasting glucose and elevated blood pressure were higher in the weight gain group than in the non-weight gain group (32% vs. 19%, *p*-value < 0.001, and 34% vs. 25%, *p*-value = 0.008, respectively). Hypertriglyceridemia was also more prevalent in the weight gain group than in the non-weight gain group (19% vs.13%, *p*-value = 0.025). The proportion of subjects with low HDL cholesterol showed no difference between the weight gain and non-weight gain groups (38% vs. 36%, *p*-value = 0.434).

### 3.3. Number of Metabolic Components in Weight Gain and Non-Weight Gain Groups

The weight gain group had a higher proportion of subjects with three or more metabolic components than the non-weight gain group (26% vs. 10%, *p* < 0.001) (Figure 1). The non-weight gain group had a higher proportion of subjects with no or one metabolic component than the weight gain group (72% vs. 54%, *p* < 0.001).

### 3.4. Body Composition in Weight Gain and Non-Weight Gain Groups

In analyzing body composition between the weight gain group and non-weight gain group by BIA, we found that the difference of muscle mass was not large (21.88 ± 4.60 kg vs. 20.7 ± 4.11 kg, *p*-value < 0.001) (Table 3). The weight gain group had a higher mean body fat mass and percentage of body fat than the non-weight gain group (20.32 ± 5.01 kg vs. 15.73 ± 5.88 kg, *p*-value < 0.001, 33.4 ± 6.53% vs. 28.88 ± 7.40%, *p*-value < 0.001, respectively).

### 3.5. Risk Factors of Metabolic Syndrome among NKRs

Table 4 shows the result of multiple logistic regression analyses to evaluate the risk of MetS among the weight gain group when compared to non-weight gain group. The weight gain group had a twofold higher risk of MetS than the non-weight gain group after adjusting for multiple sociodemographic variables and current BMI (OR, 1.875; 95% CI, 1.013–3.468). Old age, female sex, and BMI were positively associated with MetS in multivariate logistic regression analysis. However, frequent alcohol drinking, current smoking, regular exercise, higher education, higher income, stay duration in transit countries, defection period, and lipid medication had no significant association with risk of MetS.

## 4. Discussion

We investigated the association between weight gain of more than 5 kg and the risk of MetS among NKRs in South Korea. More than 5 kg weight gain after defection from North Korea was associated with a twofold higher risk of MetS after adjusting for BMI. A previous prospective study has reported a strong relationship between weight gain and the risk of MetS in Caucasian and African American populations (MetS defined according to ATP III guideline) [8]. A cross-sectional study of the Japanese adult population revealed that not only obese but also non-obese individuals who gained weight more than 10 kg over 20 years had a fivefold higher risk of MetS (defined as abdominal obesity plus one of three risk factors, high blood pressure, hypertriglyceridemia or low HDL cholesterol, high fasting glucose) [11]. In addition to long-term weight gain, the prospective study has suggested that 1 year weight gain also increases the risk of metabolic components such as abdominal circumference, blood pressure, and serum triglyceride level [5]. Another prospective study for 4 years showed that weight gain more than 2 kg was related to increased risk of hypertension and hypercholesterolemia [31]. Most studies investigating the relationship between weight gain and metabolic health have been conducted in the general population, and a few studies have evaluated the relationship between weight change and MetS among immigrants [16]. As immigrants moved from their home country to the host country, they were likely to experience changes in dietary and lifestyle habits, resulting in a significant change in body weight [12,13]. A study of Asian immigrant women who migrated to Korea suggested that those who gained weight had an increased risk of MetS (defined as obesity (BMI ≥ 25 kg/m^2^) plus more than two of the following criteria: high blood pressure, high fasting glucose, hypertriglyceridemia, or low HDL cholesterol), although no quantitative analysis was conducted [16].

Previous studies have suggested that weight gain not only increased the risk of MetS, but also increased the risk of each component of MetS. In our study, the weight gain group had higher fasting glucose than the non-weight gain group. According to 15 year follow-up prospective study, subjects with more than 5 pounds weight gain had higher fasting glucose than the weight-maintaining group [32]. A randomized controlled trial for prevention of weight gain suggested that those who gained body weight 5% or more over 6 years had higher fasting glucose levels than those who maintained or lost body weight [33]. A retrospective study of the European population also supported the hypothesis that a change in BMI during early adulthood was associated with increased HbA1C [9]. However, another cross-sectional study showed that weight gain history was not associated with high fasting glucose [34].

Our study suggested significant positive relationship between weight gain and hypertriglyceridemia, which is consistent with previous studies [6,32,34]. A prospective study of the Caucasian and African American populations reported that normal-weight Caucasian men experienced a 16% increase in triglyceride level for each 9.1 kg of weight gain over 10 years [6]. Another cross-sectional study also suggested that weight gain during adulthood was related to high triglyceride level after controlling for current BMI [34].

There is controversy over the relationship between weight gain and blood pressure [6,34,35]. Prospective studies over 10 years showed that an average weight gain of 9.1 kg was related to 2.7–3.6 mmHg increase in blood pressure [6]. In contrast, another cross-sectional study found that weight history was not associated with risk of hypertension [34]. A longitudinal study over 5 years also observed no significant association between weight gain history and systolic or diastolic blood pressure in overweight African American population [35]. Our study found positive association between weight gain and high blood pressure among NKRs in South Korea. Differences in length of follow-up, in levels on baseline BMI, or racial differences could influence the association between weight gain and blood pressure.

A significant association between weight gain and decrease in HDL cholesterol has been reported consistently in previous studies [6,32,33,34]. A prospective study suggested weight gain over 10 years was associated with decrease in HDL cholesterol from 0.09 mmol/L (Caucasian women) to 0.11 mmol/L (Caucasian men) [6]. Another prospective study over 15 years showed that weight gain was associated with decrease in HDL cholesterol level of 3.6 mg/dL (normal weight Caucasian population) to 3.8 mg/dL (overweight Caucasian population) [32]. A randomized controlled study for prevention of weight gain found that weight gain more than 5% had an adverse effect on HDL cholesterol [33]. Another cross-sectional study showed that weight gain in adulthood increased risk of low HDL cholesterol and high triglyceride [34]. However, we found no significant relationship between weight gain and low HDL cholesterol among NKRs in Korea. The differences between previous studies and our study might be due to racial differences, and further longitudinal studies are needed to clarify what has caused the differences of results between studies.

Although the weight gain group had higher BMI than the non-weight gain group on examination, the weight gain group had been thinner than the non-weight gain group in North Korea. One possible explanation for this is that NKRs who experienced severe food shortage in North Korea might gain weight due to relative abundance of food in transit countries and South Korea [19,22]. NKRs who experienced food shortage and severe malnutrition in the early life could benefit from increasing body muscle and fat mass after defection from North Korea. However, the thrifty phenotype hypothesis proposed poor nutrition in early life could produce altered glucose–insulin metabolism, which increases susceptibility to development of type 2 diabetes mellitus [36,37]. Furthermore, the weight gain group had a higher fat mass and percentage of body fat than the non-weight gain group on examination, suggesting that weight gain in the weight gain group after defection from North Korea may have led to increase in fat mass rather than muscle mass [38]. Studies have suggested that low muscle and high fat mass is related to insulin resistance and the risk of MetS [39,40,41]. Kim et al. demonstrated that higher muscle and lower fat mass were related to a lower insulin resistance and lower muscle and higher fat mass were associated with a higher insulin resistance and the risk of MetS (defined by modified NCEP-ATP III guideline) in the Korean population [39]. Srikanthan et al. also found that subjects in the highest quartile of skeletal muscle mass had a lower HOMA–IR value compared to those in the lowest quartile [40]. An increase in fat mass followed by weight gain over a relatively short time during resettlement in South Korea via transit countries could have increased insulin resistance and the risk of MetS among weight gain group in our study.

This study had several limitations. First, scientific sampling method was not applied in the NORNS study. Random sampling of NKRs in South Korea is difficult because only a few NKRs live in South Korea and they tend not to reveal their identities due to personal threats. However, the general demographics of participants in the NORNS study were similar to those of NKRs in South Korea who participated in a large-scale national survey in terms of gender ratio, age distribution, place of birth and stay duration, in transit countries [42]. Second, because of the cross-sectional design, no causal relationship between weight gain and risk of MetS could be determined. In addition, it was difficult to distinguish whether the difference in body composition between the weight gain group and the non-weight gain group was determined before migration or during migration and resettlement in South Korea. Third, this study used a self-reported questionnaire to measure weight change in NKRs subsequent to their defection from North Korea, although the validity and accuracy of self-reported measures of weight and weight change have been established in previous studies [43,44]. Fourth, NKRs have been in transit countries for different periods, which might have affected the risk of MetS. Fifth, we used BIA for body composition analysis. Although computed tomography (CT) or dual-energy X-ray absorptiometry (DXA) are the gold standard for evaluating distribution of body fat, BIA, which is highly correlated with CT and DXA, is the most frequently used method because of its simplicity, convenience, and low cost. [45,46]. Lastly, we did not implement a diet questionnaire in our survey, although there is a strong relationship between dietary patterns and MetS [47]. In the phase 2 study, we are collecting data by incorporating a diet questionnaire. In near future, our follow-up study will be able to elucidate the relationship between food consumption pattern and the risk of MetS in NKRs. We might better understand the effect of individual food and nutrients, such as whole-grain or soybean, on the risk of MetS [48,49]. Investigating relationship between changes in dietary pattern and MetS in NKRs will exhort to evaluate the changes in their microbiome among NKRs [50].

## 5. Conclusions

Our study suggested that the weight gain in NKRs during resettlement in South Korea increased the risk of MetS (defined as more than three out of five criteria: central obesity, impaired fasting glucose, elevated blood pressure, hypertriglyceridemia, and low HDL cholesterol) after adjusting for current BMI. An increase in body fat mass due to excess weight gain among NKRs might have an adverse effect on their metabolic health. Further prospective studies are needed to address the change in body weight of NKRs and its effect on the risk of cardiovascular disease and type 2 diabetes mellitus.

## Figures and Tables

**Figure 1 ijerph-18-08479-f001:**
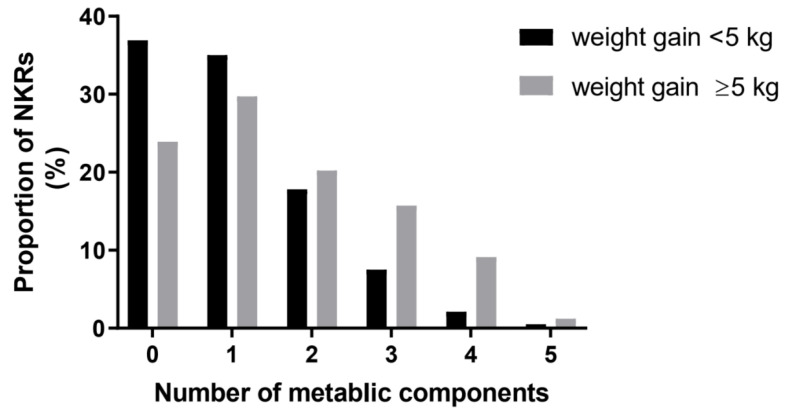
Number of metabolic syndrome components according to degree of weight gain.

**Table 1 ijerph-18-08479-t001:** Characteristics of North Korean refugees according to degree of weight gain.

	Weigh Gain < 5 kg (*n* = 546)	Weight Gain ≥ 5 kg (*n* = 253)	*p*-Value
Age	43.25 ± 12.31	44.78 ± 12.97	0.108
Sex (male)	109/545 (20%)	53/253 (21%)	0.794
Height (cm)	156.26 ± 7.28	157.3 ± 7.41	0.061
Body weight (kg)	53.60 ± 7.37	60.7 ± 8.65	<0.001
BMI (kg/m^2^)	21.98 ± 2.47	24.48 ± 2.68	<0.001
Waist circumference (cm)	76.50 ± 7.79	83.69 ± 8.57	<0.001
Fasting glucose (mg/dL)	93.21 ± 10.76	95.93 ± 13.64	0.003
Total cholesterol (mg/dL)	174.93 ± 37.21	178.85 ± 38.60	0.173
Triglyceride (mg/dL	97.61 ± 92.92	113.04 ± 78.62	0.023
HDL cholesterol (mg/dL)	54.86 ± 28.58	51.38 ± 12.16	0.065
AST (mg/dL)	21.42 ± 11.45	22.22 ± 14.73	0.407
ALT (mg/dL)	18.01 ± 17.65	21.75 ± 23.08	0.012
HOMA-IR	1.38 ± 1.29	1.68 ± 1.11	0.002
HOMA-ß	74.06 ± 45.76	79.67 ± 44.90	0.112
Systolic BP (mmHg)	116.81 ± 16.71	121.69 ± 18.30	<0.001
Diastolic BP (mmHg)	75.27 ± 12.80	77.79 ± 12.76	0.01
Current smoker	65/546 (12%)	23/253 (9%)	0.238
Frequent alcohol drinking *	349/546 (64%)	156/253 (62%)	0.506
Regular exercise ^†^	262/546 (48%)	134/253 (53%)	0.191
BMI in North Korea (kg/m^2^)	22.14 ± 2.69	20.46 ± 2.71	<0.001
Weight in North Korea (kg)	54.10 ± 7.78	50.74 ± 8.32	<0.001
Weight change (kg)	−0.5 ± 3.96	10.0 ± 4.8	<0.001
Defection period (yr)	6.67 ± 4.62	7.91 ± 4.97	0.001
Stay in transit country (yr)	3.58 ± 3.76	3.86 ± 3.68	0.333
Income (>10^6^ KRW/month)	95.49 ± 93.73	108.15 ± 132.28	0.196
Higher education ^‡^	115/546 (21%)	48/253 (19%)	0.582

Categorical variables are shown as number (%); continuous variables are given as mean ± standard deviation. * Frequent alcohol drinking’: more than one bottle of alcohol per week. ^†^ Regular exercise: vigorous activity more than one hour per week. ^‡^ Higher education: more than college graduate. Abbreviations: BMI, body mass index; HDL cholesterol, high density lipoprotein cholesterol; HOMA-IR, homeostatic model assessment for insulin resistance; HOMA- ß, homeostatic model assessment for beta cell function; BP, blood pressure; KRW, Korean won.

**Table 2 ijerph-18-08479-t002:** Comparison of metabolic syndrome and its components according to degree of weight gain.

	Weigh Gain < 5 kg (*n* = 543)	Weight Gain ≥ 5 kg (*n* = 250)	*p*-Value
Metabolic syndrome	53/527 (10%)	63/243 (26%)	<0.001
Central obesity *	64/537 (12%)	89/247 (36%)	<0.001
Impaired fasting glucose ^†^	103/543 (19%)	80/250 (32%)	<0.001
Elevated blood pressure ^‡^	134/538 (25%)	85/250 (34%)	0.008
Hypertriglyceridemia ^§^	72/543 (13%)	47/250 (19%)	0.025
Low HDL cholesterol ^‖^	195/543 (36%)	95/250 (38%)	0.434

* Central obesity, ≥90 cm (men), ≥85 cm(women); ^†^ impaired fasting glucose, fasting glucose ≥100mg/dL; ^‡^ elevated blood pressure, systolic BP ≥130mmHg or diastolic BP ≥ 85 mmHg; ^§^ hypertriglyceridemia, triglyceride ≥150 mg/dL; ^‖^ low HDL cholesterol <40mg/dL (men), <50 mg/dL (women).

**Table 3 ijerph-18-08479-t003:** Comparison of body composition according to degree of weight gain.

	Weigh Gain < 5 kg (*n* = 545)	Weight Gain ≥ 5 kg (*n* = 252)	*p*-Value
Body weight (kg)	53.60 ± 7.37	60.7 ± 8.65	<0.001
Muscle mass (kg)	20.67 ± 4.11	21.88 ± 4.60	<0.001
Fat mass (kg)	15.73 ± 5.88	20.32 ± 5.01	<0.001
Percentage of body fat (%)	28.88 ± 7.40	33.4 ± 6.53	<0.001

Variables are given as mean ± standard deviation.

**Table 4 ijerph-18-08479-t004:** Odds ratio of metabolic syndrome risk among North Korean refugees.

	Model 1 (*n* = 769)		Model 2 (*n* = 509)	
	OR (95% CI)	*p*-Value	OR (95% CI)	*p*-Value
Age *	1.073	< 0.001	1.056	<0.001
	(1.049–1.097)		(1.030–1.081)	
Sex	2.708	0.032	2.852	0.029
(male as reference)	(1.090–6.729)		(1.112–7.317)	
Weight gain (≥5 kg)	3.392	< 0.001	1.875	0.045
	(1.973–5.832)		(1.013–3.468)	
Current smoker	2.479	0.100	3.128	0.051
(nonsmoker as reference)	(0.841–7.307)		(0.997–9.820)	
Frequent alcohol drinking ^†^	0.962	0.887	0.937	0.827
(nondrinker as reference)	(0.560–1.62)		(0.520–1.686)	
Regular exercise ^‡^	0.818	0.468	0.724	0.290
	(0.476–1.407)		(0.398–1.317)	
Stay in transit country (years)	0.973	0.629	0.988	0.853
	(0.870–1.088)		(0.873–1.119)	
Defection period (years)	1.010	0.817	0.979	0.644
	(0.930–1.096)		(0.893–1.072)	
Higher education ^§^	0.843	0.612	0.728	0.393
	(0.435–1.633)		(0.351–1.509)	
Higher income	0.579	0.094	0.634	0.190
(>10^6^ KRW/month)	(0.305–1.098)		(0.321–1.254)	
BMI			1.312	0.001
			(1.177–1.463)	
Lipid medication			0.866	0.480
			(0.651–0.125)	

* Age as continuous variable; ^† ‘^frequent alcohol drinking’: more than one bottle of alcohol per week; ^‡^ regular exercise: vigorous activity more than one hour per week; ^§^ higher education: more than college graduate; abbreviations: OR, odds ratio; CI, confidence interval; BMI, body mass index; KRW, Korean won. Model 1, adjusted by age, sex, current smoking, alcohol drinking, regular exercise, stay period in transit countries, defection period, education, income. Model 2, adjusted by Model 1, lipid medication, and BMI.

## Data Availability

The data presented in this study are available on request from the corresponding author.

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
