# Peer review of "Weight Gain Predicts Metabolic Syndrome among North Korean Refugees in South Korea"

_ijerph, 2021, doi:10.3390/ijerph18168479_

Round 1

Reviewer 1 Report

In the present paper, Yoon Jung Kim and coworkers investigated the association between weight gain during the resettlement and risk of metabolic syndrome (MetS) among North Korean refugees (NKRs) in South Korea. The authors concluded that the weight gain in NKRs during resettlement in South Korea increased the risk of MetS after adjusting for current BMI. Weight gain group had a higher percentage of body fat than non-weight gain group, which may have had an adverse effect on their metabolic health. Overall, I think that the paper could be of interest to the readers and researchers, in general.

I make some suggestions for further improve the quality of the manuscript.

1) The authors honestly acknowledged that further prospective studies are needed to address the change in body weight of NKRs and its effect on the risk of cardiovascular disease and type 2 diabetes mellitus among NKRs in South Korea. Do the authors think it possible to predict results of next prospective studies, based on the present data? I mean, based on the trends observed in this study, what is justified to expect on other indicators of MetS?

2) The authors, if possible, should incorporate in tables the dietary pattern of the patients included in the present study (e. Mediterranean-style diet, Plants-based diet, Nordic dietary pattern, etc.); in this way, I feel that the readers can better understand the results obtained in the present clinical study and their possible application to clinical practice.

3) “Oriental diet” is particularly abundant in isoflavones. So, the overall effects on MetS observed in clinical studies could be related to these phytochemicals and no adherence/not compliance to diet. Please discuss this crucial aspect in the revised manuscript.

4) Some medicines/drugs can be used to treat MetS. This aspect could interfere with results here revealed. Please discuss this aspect and eventually take this factor into account in multiple logistic regression analysis.

5) Recent research suggested that dietary fibers from beans, fruits and vegetables were associated with the gut microbiome composition and, accordingly, with risk of developinh MetS. Does the authors plan to assess microbioma composition in their population? Please make a comment in the discussion section of revised manuscript.

Reviewer 2 Report

Abstract

  • This section should start with some brief background information.
  • The study design, the age of participants and the statistical analysis used should be clearly mentioned.
  • OR and P-value should be stated only. Please delete 95% CI.

Introduction

  • Line 41: obesity and weight gain are similar terms. Please delete weight gain.
  • Line 42: Do you mean "general adult population".
  • Line 42,44,45,48,54,56: Please define the study design of these studies (i.e., cross-sectional, longitudinal…etc).
  • Please change weight gain to obesity (Line 42,45,49,50,65).
  • Line 42-43: I would suggest including the mechanisms which have been forward to support relationships between obesity and MetS in adults.
  • Line 52-54: Please define these countries. Are you referring to Asian countries?
  • Line 56-57: More details are needed about the result of these studies.
  • Please provide evidence about the prevalence of obesity/MetS among Northern and Southern Korean adults.
  • No studies from international contexts were used to tightly support the research aim.
  • Line 70: Please define MetS here (i.e., triglyceride, HDL,LDL…etc).
  • Please provide the research hypothesis at the end of the section.

Methods

  • Line 75-76: Please define the study design here.
  • Line 78-79: Please describe how the participants were recruited?
  • The inclusion criteria are not clearly defined. A diagram is needed to show the final selection of participants.
  • Line 88: The age of participants should be reported here.
  • Line 94-101: Additional information is required of all socio-demographic variables. Please also provide information about the instrument/questionnaires used to collect these variables. Were they valid/reliable?
  • Line 103-107: Reference(s) should be included here.
  • More details are needed on how weight, height, systolic and diastolic blood pressure. For example, by whom were these measured?
  • Sections 2.2.3 and 2.2.4 should be merged.
  • Data collection procedure should be clearly described.
  • Line 129: Clear rationale why NKRs classified into weight gain groups (± 5 kg)?
  • Line 133-139: Statistical analysis should be described in more details.

Results

  • Line 180: "The mean number of metabolic components", meaning unclear, please clarify.
  • Line 198-203: All results whether significant (OR, P-value) or not should be reported.
  • Line 205-206: This should be moved to statistical analysis section and described in details.
  • Table 4: Why females were identified as a reference? Why weight gain (≥ 5kg) was included? Age, regular exercise and higher education should be clearly defined

Discussion

  • Line 210-211: "Studies"-please refer to my previous comments in introduction. The study design should be clearly reported.
  • Line 212: "Asian adult population"
  • Line 214,215,216,217,221,267,270: Please define MetS reported in these studies.
  • Line 223, Line 232-233: These statements do not add any meaning. Please expand.
  • Line 232-258: I found it difficult to develop a clear message from these paragraphs. Please discuss these results thoroughly and not just refer to previous studies.
  • Line 261-266: meaning unclear-please clarify by adding references.
  • Line 296: Please define MetS here.
  • Line 295-300: The implications of the study should be clearly reported.

Round 2

Reviewer 1 Report

The authors have satisfactorily responded to all my questions and made the necessary changes to the manuscript.

Reviewer 2 Report

No further comments.